# *Ornithobacterium rhinotracheale*: An Update Review about An Emerging Poultry Pathogen

**DOI:** 10.3390/vetsci7010003

**Published:** 2019-12-27

**Authors:** Eunice Ventura Barbosa, Clarissa Varajão Cardoso, Rita de Cássia Figueira Silva, Aloysio de Mello Figueiredo Cerqueira, Maíra Halfen Teixeira Liberal, Helena Carla Castro

**Affiliations:** 1Programa de Pós-Graduação em Ciências e Biotecnologia, Universidade Federal Fluminense, Niterói 24210-130, Brazil; euniceventurabarbosa@hotmail.com (E.V.B.); cacavc@terra.com.br (C.V.C.); 2Centro Estadual de Pesquisa em Sanidade Animal—CEPGM, Empresa de Pesquisa Agropecuária do Estado do Rio de Janeiro, Niterói 24120-191, Brazil; rcassiasilva@gmail.com (R.d.C.F.S.); mairahalfen@gmail.com (M.H.T.L.); 3Departamento de Microbiologia e Parasitologia (MIP), Universidade Federal Fluminense, Niterói 24210-130, Brazil; cerqueira.aloysio@gmail.com

**Keywords:** respiratory disease, poultry health, ornitobacteriosis, ORT

## Abstract

Respiratory diseases in birds generate sanitary and economic impacts and may be related to the environment and climate. *Mycoplasma gallisepticum*, *Ornithobacterium rhinotracheale* (ORT), *Pasteurella multocida*, *Avibacterium paragallinarum*, *Escherichia coli*, *Riemerella anatipestifer*, and *Bordetella avium* are among the most important avian respiratory pathogens. ORT is responsible for causing ornitobacteriosis, a disease characterized by clinical signs ranging from mild to severe respiratory conditions, with high mortality rates, mainly affecting turkeys and chickens. The first report of ornitobacteriosis was in 1981 in Germany. Despite its importance, few studies on ORT have been published. In addition, the presence of this pathogen has been neglected in poultry farms, mainly due to the lack of appropriate diagnostic protocols. The lack of correct isolation and diagnostic protocols along with inappropriate use of antimicrobial agents have been contributing to treatment failure. Due to its economic importance to the poultry industry, ornitobacteriosis should be monitored and included in national programs for the prevention and control of avian respiratory diseases. This review aimed to update and discuss important issues related to ORT since this pathogen has great economic and sanitary implications for the chicken production chain.

## 1. Introduction

Respiratory diseases in poultry are mostly accompanied by heavy economic losses and increased mortality rates, medication costs, and condemnation rates due to aerosacculitis and decreases in egg production. These infections have been recognized in many countries worldwide [1,2,3,4] and are associated with several microorganisms, including viruses and bacteria, and influenced by climate changes [5].

The high-density confinement of birds causes behavioral problems and physical injury to animals. It is also related to stress in birds due to movement restriction and the unhealthy conditions generated by crowding, leading to a reduced air quality, which contributes to the onset of respiratory tract diseases [6,7]. Chicken meat is the second most consumed meat and represents approximately 36% of world meat production. Therefore, poultry respiratory diseases are not only a problem for the biggest producers, but also a global concern [8,9].

The most important etiological agents of poultry respiratory diseases include: (a) viruses (Newcastle disease, avian metapneumovirus, avian infectious bronchitis); (b) fungi, such as *Aspergillus* spp.; and (c) bacteria, including *Mycoplasma gallisepticum*, *Pasteurella multocida*, *Avibacterium paragallinarum*, *Escherichia coli*, *Riemerella anatipestifer*, *Bordetella avium*, and *Ornithobacterium rhinotracheale* [9,10].

*Ornithobacterium rhinotracheale* (ORT) causes ornithobacteriosis, a contagious disease transmitted horizontally by direct contact, aerosols, or indirectly through drinking water [5,11]. Vertical transmission is still unclear, but probable [12]. According to the World Organization for Animal Health—OIE [13] (Section 2: Terrestrial Animal Health Code), ORT is a threatening, but not zoonotic microorganism [14,15,16].

This review aimed to discuss pathogenic infections in poultry farms caused by this agent, emphasizing the clinical, bacteriological, and genetic characteristics of pathogenic strains. We also addressed the importance of the host in the pathogenesis of infection, as well as poultry as a dispersion factor and the emergence of antimicrobial resistance.

## 2. Ornithobacterium rhinotracheale (ORT)

ORT is a gram-negative, non-mobile, non-sporulating bacterium [12], belonging to the superfamily V rRNA and family Flavobacteriaceae. It is from the *Cytophaga-Flavobacterium*-*Bacteroides* descending genetic line. ORT was previously designated as gram-negative pleomorphic rod [17], *Pasteurella*-like [18], *Kingella*-like [19], or Taxon 28 [18].

Charlton et al. [17] did the preliminary characterization of a pleomorphic gram-negative rod associated with avian respiratory disease, followed by a genotypic classification by Vandamme et al. [20] as a new genus and species. Currently, 18 ORT serotypes (A–R) have been reported, with no direct relationship with virulence. Due to the difficulty of the serotyping process, it has been suggested that there are new serotypes yet to be determined [2,3,21].

Although described as a non-motile, non-hemolytic bacterium with inconsistent biochemical properties, ORT strains presenting a β-hemolytic phenotype were isolated from the lungs and tracheas of chickens with pneumonia in Argentina [22].

## 3. Clinical and Pathologic Features

### 3.1. Mode of Infection and Transmission

ORT is considered to be an emergent poultry breeding respiratory pathogen, acting, in most cases, opportunistically, although it may be the primary cause of an infection [23]. The disease is usually associated with respiratory symptoms, growth retardation, reduction of egg production, and mortality, with economic losses to the poultry industry [12]. The respiratory symptoms include tracheitis, aerosacculitis, pericarditis, sinusitis, and exudative pneumonia, with fibrin purulent lesions and often unilateral pneumonia [23].

The severity of ORT clinical disease, mortality rate, and decreased productivity are variable. Factors, such as management, environmental stressors, or the presence of other pathogens, can play an important role in the disease manifestation [12]. Gavrilović et al. [24] have demonstrated that natural monoinfections with ORT are less aggressive than those from experimental challenge. Experimental studies of monoinfection with ORT showed that ORT had less effect on hens than on pheasants, suggesting it is host dependent, although both species belonged to the same Phasianinae subfamily.

Additionally, the lesions generated in both species are dissimilar to ORT infection in turkeys and this host belongs to another subfamily (*Meleagridinae*) [25]. In turkeys, a flattened tracheal mucosa, reddish or hemorrhagic spots, and accumulation of mucus in pathological lesions are common. The lungs may have hemorrhagic lesions, in which blood is eliminated through the mouth [26,27]. Erythrocytes in the airways caused by ORT infection lead to hyperemia with subsequent blood congestion [28].

### 3.2. Epidemiology and Prevention

There are several molecular studies on wild birds transmitting respiratory diseases to poultry, representing the initial route of the disease [5,23]. Several ORT isolates have been identified from birds, including chicken, turkey, duck, goose, pigeon, quail, ostrich, seagull, partridge, and pheasant [14]. ORT incidence is higher in turkeys, with an isolation frequency of 41% compared to 6.9% of broiler chickens [27]. The presence of different serotypes (A, B, C, D, and E) with variable adherence profiles suggests that these serotypes have different virulence factors [29,30,31,32]. According to Sharifzadeh and colleagues, toxic activities, fimbriae, pili, or even relevant plasmids have not been described [33]. Walters has suggested that an ORT hemolytic isolate had a 4-kb plasmid similar to that found in *Riemerella anatipestifer* [34].

Associations with other microorganisms is rare, but in the case of the protozoan *Cryptosporidium* spp., contribute to an immunosuppressive effect, the occurrence of ORT as a secondary infection is increased. The full understanding of the synergistic role of these microorganisms remains to be clarified, but it is known that the association of these pathogens is more serious than the pathogen alone [4,22,27,35,36,37].

Poultry respiratory diseases involving ORT have been reported worldwide. In Cuba, for instance, it has been reported in laying hens with chronic respiratory syndrome [38]. In 2015, New Zealand reported the first case of ORT in broiler chickens. The suspected birds were subjected to diagnostic tests as part of an investigation by the Ministry of Industries, with protocols standardized in an animal laboratory in environments of physical containment level 3 [39].

In Brazil, reports on the prevalence and identification of ORT are rare. However, in 1998, the presence of antibodies in poultry breeding was detected in the states of São Paulo and Minas Gerais. In 2001, the first isolation of ORT was made in Rio Grande do Sul state, reinforcing the idea of circulation of the pathogen. This is expected, since Brazil’s border countries have a high index of ORT isolation [40].

Umali and colleagues [41] in Japan, demonstrated the ability of ORT to cause systemic disease in broiler chickens where ORT was isolated in blood samples from the heart, liver, kidney, spleen, and ovaries. The authors affirmed the need for further studies to determine the potential relevance of the association with other pathogens.

To be considered a good animal breed, it is necessary to have a good genetic line, with appropriate management as well as protocols for the prevention and control of infectious diseases [42]. Unfortunately, the indiscriminate use of antimicrobial agents in rural environments, sometimes without proper prescription by a veterinarian, can contribute to the generation of multidrug-resistant strains of some bacteria in animal farms [43].

Since the ORT infection has become endemic, controlling the disease during poultry breeding is important. Thus, good health and prophylactic care are recommended, following the principles of biosafety, one of which has been the use of the “all-in/all-out” poultry industry [44]. This system is characterized by the acquisition of a group of chicks in the aviary where they are raised and then slaughtered. The time for cleaning and disinfection must be respected to prevent the spread of microorganisms until a new group arrives. The use of aldehyde-based chemicals or organic acid disinfectants can completely inactivate the ORT and is highly effective even at low concentrations and contact times [44].

The incorrect diagnosis of the microbial agent and the lack of its antimicrobial susceptibility profile further worsen the inefficient use of antimicrobial agents against respiratory diseases in birds. This contributes to the generation of more resistant variant strains that will contaminate the soil and streams, spreading to other animals and humans [45].

### 3.3. Clinical Signs and Pathological Lesions

In the pathogenesis of ORT-associated infection, the severity of the signs and mortality rates vary and are influenced by several environmental factors (e.g., high levels of ammonia, inadequate room ventilation, extremes of temperature and humidity, and high storage densities) [12]. In a recent experimental study, Ellakany et al. [46] confirmed the respiratory signs and histopathological changes caused by ORT. The authors also verified that infection associated with *Mycoplasma gallisepticum* increased the severity of respiratory symptoms, such as pneumonia, edema, and severe infiltration of inflammatory cells into the lungs and air sacs. In addition, there was a negative effect on body weight and performance of birds.

Viral infections often cause acute respiratory problems from which birds can usually recover quite easily. When bacterial pathogens are involved, however, the problem is more critical, especially when bacterial synergism occurs. Such synergistic bacterial species include *Escherichia coli*, *Pasteurella multocida*, *Bordetella avium*, *Ornithobacterium rhinotracheale*, *Mycoplasma gallisepticum*, *M. synoviae*, *M. iowae*, *M. meleagridis*, *M. imitans*, *Chlamydophila psittaci*, and *Riemerella anatipestifer*. In general, ORT lesions are nonspecific and easily misidentified as other respiratory infections, but when several pathogens are associated, the lesions become more severe, often leading to death [12,47].

Pan et al. [4] studied the pathogens responsible for the progressive pneumonia and reported the first experimental co-infection in broiler chickens, associating ORT with H9N2 avian influenza virus (AIV). They demonstrated that ORT alone could induce a high mortality rate, with around 50% lethality. In addition, ORT infection led to higher mortality if H9N2 AIV was also present, with lethality of 70% and 90%, respectively, if ORT inoculation was simultaneously made with H9N2 AIV or if H9N2 AIV was inoculated after 3 days.

After an outbreak of bronchial embolization in broilers in China, strains of ORT and *Streptococcus zooepidemicus* were isolated. An experimental study with these strains confirmed that ORT infection alone could induce infection and cause high mortality and co-infection with *S. zooepidemicus* was even more lethal, indicating that polymicrobial infections may be related to the outbreak of bronchial embolization [48].

The ORT’s pathogenesis severity is associated with the environments factors, biofilm formation, and synergism with others pathogens, contributing to the persistence of the microorganism (Figure 1) [49].

The production (expression) of a hemagglutinin and neuraminidase have been described in ORT [50]. These virulence factors may play an important role in tissue colonization and inflammation in the host [51]. In addition, although ORT hemolysin may also act as a virulence factor, recent studies have shown that non-hemolytic isolates can survive longer in the host and present high pathogenicity [52].

Adherence is an important step in bacterial pathogenesis. The mechanism of ORT adhesion to host tissues is unknown, but hemagglutinins and other glycoproteins probably mediate this step. Cell viability studies have shown that ORT can adhere to chicken embryo lung cells after 3 h incubation, resulting in a diffuse growth pattern. There are appendages and hemagglutinins, which probably contributed to the adhesion, suggesting the presence of host cell receptors that recognize ORT adhesins [21].

## 4. Laboratory Diagnosis

### 4.1. Isolation and Identification

As clinical signs associated with ORT infections are variable and nonspecific, it is very important to have a specific and sensitive laboratory diagnostic test [12]. Phenotypic, immunologic, and/or molecular diagnoses still need to be developed or improved. As respiratory injuries associated with ORT infections may be caused by several other bacteria, including *E. coli*, *Avibacterium paragallinarum*, and *Pasteurella multocida,* selective and specific assays are required (Table 1) [5,12,53].

ORT is a fastidious bacterium and needs good supplementation and proper conditions for culturing. The addition of antimicrobial agents (gentamicin or polymyxin) to the culture medium is required, since other bacteria, such as, *E. coli*, *Proteus* spp., *Pseudomonas* spp., can mask ORT growth. For cultivation, blood agar, Columbia agar, or soybean casein agar, supplemented with 5% to 10% sheep blood and 10 μg/mL gentamicin or polymyxin, are used. ORT usually forms small, circular, white to gray colonies [18,53]. Currently, no ORT selective and/or specific culture medium is commercially available [54]. The incubation is typically performed under anaerobic or microaerophilic conditions with 7% CO_2_ at 37 °C for 24 to 48 h [18,20,54], which hinder the process and increase costs. Under these conditions, highly pleomorphic rods are observed by microscopy [12].

### 4.2. Serotyping and Detection

Many researchers have used serology to investigate ORT infections, but the results show wide variations. In the poultry industry, the presence of anti-ORT antibodies in chickens has been reported in many countries [2,15,22,24,27,28,36,37,41,52,55,56,57]. Serology has advantages compared to bacterial isolation, since antibodies persist for several weeks after infection [5,12,53]. According to the literature, more than 18 serotypes (A–R), have been described. Serotype A is the most prevalent one in chickens (94%) and turkeys (57%) [58,59].

An indirect ELISA test developed to use in chickens (IDEXX ORT Ab Test) was also used since there could be a cross reaction with antibodies produced after infection. Of note, chickens and pheasants belong to the same family. The results showed a high similarity in the kinetics of serological responses for both species after infection, also confirming the use of this test in pheasants [24].

The serum plate agglutination test (SPAT), also known as the rapid serum agglutination test (RAT), aims to monitor several diseases, including ORT, especially in industrial poultry. It has the advantages of being practical, low cost, and easy to use in the field. These features contribute to the speed of the results and the ability to take rapid action. In comparison to the RAT, the ELISA test requires a longer time and cost for its execution [60].

In reports on ORT isolation and detection of antibodies in laying hens by RAT, a cross-reaction was observed with serotype B and serotype A and between serotypes I and L. Other studies showed low specificity (e.g., cross-reactions) detected for serotypes A, E, and I, but not with serotype C, which highlights certain disadvantages of this methodology [12,61].

The collection of samples for ORT isolation with dry sterile swabs and maintenance at 4 °C for 2 days or −20 °C for 5 days retained the bacterial viability and hindered the growth of other bacteria [62]. For biochemical identification, some commercial diagnostic kits are also available (e.g., API-20NE and API-ZYM), which have been tested and adapted for ORT identification [12].

The API-20NE kit (bioMérieux, Marcy L’Etoile, France) is intended for the identification of non-fastidious gram-negative rods not belonging to the *Enterobacteriaceae*. It is interpreted through a numerical system and has an ORT-adapted reading using at 30 °C. Almost all strains (99%) have a bio code of 0-2-2-0-0-0-4 (65%) or 0-0-2-0-0-0-4 (34%) [12,63,64]. The API-ZYM kit (bioMérieux, Marcy L’Etoile, France), based on semiquantitative enzymatic activities, with an adapted interpretation for ORT through the negative reaction of five enzymes (lipase, β-glucosidase, β-glucuronidase, α-fucosidase, and α-mannosidase) [12,65,66,67].

## 5. Genetic Relatedness

The complete genome and sequencing data from the ORT-UMN 88 and ORT strains H06-030791 contributed to the understanding of the genome evolution, virulence factors, and pathogenesis profile. ORT-UMN 88 was isolated from turkey pneumonia in 1995 and was the first to be used in infection reproduction experiments. This strain depends on high levels of iron for growth and also has β-hemolytic activity. The hemolysin protein acts as a virulence factor whereas the presence of mobile elements can transfer resistance to other bacteria. Interestingly, there are small portions of DNA with nucleotide repetitions, called clustered regularly interspaced short palindromic repeats (CRISPR), which seem to be involved in the strain’s natural defense. In contrast, ORT H06-030791 was isolated from turkey lungs in 2006 and demonstrated strong β-hemolytic activity and CRISPR, but had no iron dependence as it is not affected by the presence of chelated iron [3,68].

According to some studies, there is a great variety of species or subspecies within the genus *Ornithobacterium*, although a high similarity in the strains studied have been observed. New phylogenetic analyses of the 16S rRNA gene allowed comparisons of isolates from different countries, showing the existence of a great variety among ORT strains, in contrast to previous impressions [69,70].

Recently, partial sequencing of the *rpo*B gene, evaluated in isolates recovered from different avian hosts, has been proposed for laboratory diagnosis [71]. The phylogenetic relationships was compared to the results of 16S rRNA gene sequencing and multilocus sequence typing (MLST) in 65 isolates. The 16S rRNA gene was found to demonstrate identity ranging from 85.1% to 100%. This sequencing protocol can identify and differentiate *Ornithobacterium* species [72]. Genetic characterization of ORT isolates can also be performed by modified PCR techniques, such as enterobacterial repetitive intergenic consensus-PCR (ERIC-PCR), repetitive element palindromic PCR (REP-PCR), random amplified polymorphic DNA-PCR (RAPD-PCR), and MLST [56].

## 6. Treatment

The use of antimicrobial agents to treat respiratory infections in poultry has often been problematic for veterinary clinics due to the high costs and difficulty of reabsorption by the intestine [73]. Additionally, the acquired antimicrobial resistance can makes antibiotic therapy an ineffective alternative for the treatment of these conditions [74]. It is important to consider that consumption of animal foods, such as chicken meat, can also serve as an antimicrobial resistance source [75].

It is known that ORT easily acquires antimicrobial resistance and there is great inconsistency about sensitivity/resistance distinction, which may depend on the phenotypic profile of the strain and the geographical origin. [69,73,76]. There is no specific standard for the ORT antimicrobial susceptibility profile and the CLSI recommendations for fastidious gram-negative microorganisms are usually applied [76,77].

Several antimicrobial agents, including those most recently developed, are becoming inefficient against ORT, reinforcing the hypothesis of continuous resistance transference among them [69,73,76], with an increased resistance for different classes of drugs [74].

Regarding the susceptibility profile of ORT, studies in Belgium have shown sensitivity to tiamulin [73], florfenicol [78], and gamithromycin [76], and resistance to ampicillin, ceftiofur, tetracycline, cotrimoxazole (sulfamethoxazole + trimethoprim), and enrofloxacin [73]. A study from Mexico has shown sensitivity to amoxicillin, enrofloxacin, and oxytetracycline, and resistance to gentamicin and fosfomycin [77].

ORT resistance to cotrimoxazole has been reported in Mexico [55], Japan [41] and Brazil [10]. Resistance to ampicillin, ceftiofur, tetracycline, cotrimoxazole, and enrofloxacin has also been observed in the Netherlands [79], the United States [80], and Hungary [81]. Although resistance to enrofloxacin and tetracycline was observed in several countries, isolates from Iran were intermediately susceptible to this agent [54] and sensitive in most countries [76]. In addition, isolates from Pakistan were susceptible to tetracycline [57].

Mutations in *gyr*A were found after experimental inoculation in turkey flocks and treatment with enrofloxacin was associated with an increase in MIC of 0.03 to 0.25 mg/mL in field isolates [82]. Back et al. [83] reported participation of plasmids in antimicrobial resistance in chicken isolates.

According to Jansen et al. [84], the presence of 14 genes (ORF1–ORF14) was observed in plasmid pOR1, encoding proteins of replication, mobilization, and resistance to heavy metals. Importantly, ORF14, with the *vap*D gene encoding a virulence-associated protein, was found in some poultry pathogens. At least 13 bacteria have this gene, including *Riemerella anatipestifer*, but its role is still unknown.

## 7. Vaccination

Vaccination could be an effective prevention/control strategy since the majority of ORT strains present resistance against the main antimicrobials used in poultry farming [85]. Thus, knowledge about the host immune response is of great importance for the development of more effective vaccines [86]. Different vaccines, including recombinant ones, have been developed and reported with variable results for the control of both natural and experimental ORT infections [74,85,87,88,89,90,91].

In 1998, van Empel and van den Bosch [87] reported the use of inactivated vaccines in commercial chickens. The application of the inactivated vaccine in poultry results in a rapid immune response, but the titer depends upon the adjuvant, which suggests that a second booster dose is required. This vaccination is more effective for poultry from 8 weeks of age, as maternal antibodies negatively affect the vaccine response by protecting the progeny up to 4 weeks of age [91].

In experimental challenges using inactivated vaccines, it was possible to obtain high maternal antibody titers in chicken matrices, resulting in sufficient progeny protection [74,87,89]. Cauwerts et al. [74] also achieved lower mortality rates and higher average production indices in progenies from vaccinated matrices. In 2007, Murthy et al. [91] produced 18 vaccines with different inactivating substances (formalin and thiomersal), with and without adjuvants (mineral oil, alum, and aluminum hydroxide gel). As a result, the group of vaccinated poultry with a bacterin adjuvant and mineral oil induced the highest serological response with a significant decrease in macroscopic lesions (e.g., pneumonia and aerosacculitis).

De Herdt et al. [92] evaluated broiler performance after vaccination with inactivated Nobilis OR (*Ornithobacterium rhinotracheale serotype A*) derivative (MSD Animal Health, Boxmeer, The Netherlands). They detected the best performance in vaccinated animals, with 22.3% reduction in broiler loss and higher production rate (3.9%). This study indicated that vaccination can improve broiler progeny performance, adding value to the poultry sector.

Generally, an attenuated vaccine is not recommended due to ORT pathogenicity. In Egypt, Ellakany et al. investigated the role of a live variant IVB 4/91 vaccine in ORT infection and revealed that its use can simultaneously increase the pathogenicity of ORT [93]. However, one study evaluated the use of a live, temperature-sensitive mutant vaccine administered by oculonasal instillation in one-day-old turkey. As a result, the strain was able to colonize the upper respiratory tract and recover 13 days after administration. The positive humoral response was only observed in the vaccinated group [88]. However, subunit vaccines have been developed to promote homologous and heterologous protection and mediate immunity without causing clinical signs [18]. The autogenic bacterins tested in turkeys were successfully used to control outbreaks in Israel [12].

Recombinant subunit vaccines have also already been developed [85,86,90]. The genes were amplified, cloned, and expressed in *E. coli*. Eight cross-reactive antigens were encoded and tested as subunits and protection in an experimental challenge formed by ORT homologous and heterologous serotypes. As a result, homologous and heterologous protection was found in the challenged chicken group, with production of reactive antibodies against recombinant proteins in Western Blot assays [86]. Although the eight antigens were not expressed by all ORT serotypes, a four-component recombinant vaccine was able to protect against the challenge in heterologous ORT infection [85].

## 8. Conclusions

ORT is a respiratory pathogen widely spread in turkey and chicken poultry farms in several parts of the world. ORT infection can be associated with other pathogens. The similarity of the clinical signs of ornitobacteriosis and other avian respiratory diseases, as well as the difficulty of carrying out a conclusive laboratory diagnosis, allows for the dissemination of ORT in the poultry production system.

Although progress on the importance of the disease in the poultry breeding has been made, there are still gaps in the knowledge, including the characterization of pathogenic isolates. Further studies are important to elucidate ORT pathogenic mechanisms, especially on the adhesion and host cells involved in the pathogenesis process.

The classical ORT identification still needs improvement. Therefore, an association with or the alternative use of molecular techniques must be considered to obtain more accurate results. Specific criteria for interpreting the ORT susceptibility profile are also needed, which will lead to better and more effective treatment schemes. Additionally, a more effective and safe vaccine is necessary.

## Figures and Tables

**Figure 1 vetsci-07-00003-f001:**
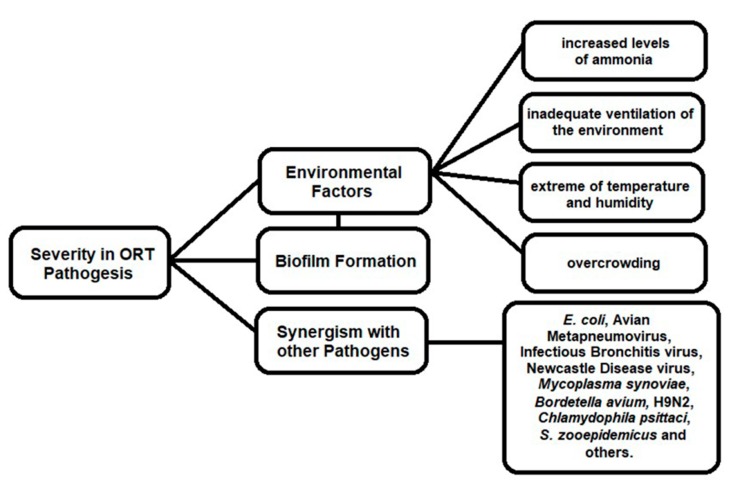
Factors involved in *Ornithobacterium rhinotracheale* (ORT) pathogenesis.

**Table 1 vetsci-07-00003-t001:** Phenotypic profile of *Ornithobacterium rhinotracheale* (ORT).

Biochemical Tests	Test Interpretation	Result
Indol *	Degradation of tryptophan yielding indole.	−
Ornithine Decarboxylase	Decarboxylation of ornithine.	−
Catalase *	Presence of catalase enzyme by decomposing H_2_O_2_ in water and O_2_.	−
Oxidase *	Detect of indophenol production (cytochrome oxidase).	+
Fructose	Sucrose fermentation producing fructose and acids.	+
Glucose	Fermentation of glucose producing acid or acid and gas.	−
Mannitol	Fermentation of mannitol with acid production during the process.	−
Maltose	Degradation of maltose generating 2 molecules of glucose.	−
Motility	Determination of bacterial motility by means of flagella.	−
Triple Sugar Iron (TSI) *	Glucose fermentation and gas production. Fermentation of lactose and/or sucrose. Production of H_2_S.	−
Growth MacConkey	Isolation of gram-negative and inhibition of gram-positive cocci.	−
Hemolysis in Blood Agar	Differentiation of hemolysis production.	−
Growth in Nutrient Agar	Medium for preliminary culture in bacteriology.	−
Voges–Proskauer	Production of acetoin from the fermentation of glucose.	+
Urease	Urease enzyme hydrolyzing urea with ammonia formation.	+
Βeta-galactosidase *	Fermentation of glucose by the enzyme beta-galactosidase. Difference lactose fermentation delay of lactose non-fermenters.	+

* Relevant reactions for ORT detection. Source: Vandamme et al. [20], Ozbey et al. [53], Chin & Charlton [11].

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
