# Peer review of "Ornithobacterium rhinotracheale: An Update Review about An Emerging Poultry Pathogen"

_vetsci, 2019, doi:10.3390/vetsci7010003_

Round 1
Reviewer 1 Report
Regarding to review entitled "Ornithobacterium rhinotracheale: a review about an important poultry pathogen", this review aimed to discuss pathogenic infections in poultry farms caused by Ornithobacterium rhinotracheale, emphasizing the clinical, bacteriological and genetic characteristics of pathogenic strains. They also addressed the importance of the host in the pathogenesis of infection, as well as poultry as a dispersion factor and the emergence of antimicrobial resistance.
The review was well written. However, there are many reviews covered the same title. Therefore, the authors must addresses the new parts covered by the current review (Novelty) in the introduction part. Additionally, I recommend to add "An updated review" in the title.
Reviewer 2 Report
Dear listed authors,
The manuscript entitled Ornithobacterium rhinotracheale: a review about an important poultry pathogen tried to summarize ORT research progress associated with pathogen biology, serotyping methods,lab diagnosis, control measures and vaccine approaches. The manuscript is interesting for potential readers to understand ORT pathogenesis and comprehensive control strategies.However, the manuscript is not qualified for the publication due to a few weakness. Please pay attention to the comments and suggestions how to improve the manuscript in the next step.
1.First of all, the title does not cover all the topic and it should be corrected, for example, Ornithobacterium rhinotracheale: a review from pathogen biology to control strategy.
2.Regarding general knowledge in Figure 3 and Figure 4, both figures should be replaced with the typical lesions of ORT infection, highlighting the difference among respiratory pathogens, for example, avian influenza virus H9N2, IBV,Chlamydia psittaci,MG, Metapneumovirus and Bordetella avium.
3. In Figure 1, ORT pathogenesis is not sufficient to address significant difference between ORT alone and ORT co-infection. Particularly, please clarify Mycoplasma synoviae or MG involves in synergism with ORT infection in Figure 1. Also,H9N2 and Streptococcus zooepidemicus were reported to involve in ORT infection and should be addressed the previous publications.
3.Both Table 2 and Table 3 are required to list advantages and disadvantages compared to the commercial assay.
4.Some descriptions need re-orgnization for better understanding, for example. Lines 314-318 should be moved to Line 298.Lines 319-332 should be moved to Line 106.
Reviewer 3 Report
The manuscript aims to give a summary of our current knowledge on Ornithobacterium rhinotracheale. Unfortunately, this goal has not been fully achieved. The manuscript is not very well structured, some chapters contain too much unnecessary or difficult to interpret information. For example, the epidemiology and prevention chapter is rather confusing and does not contain any relevant statement. In addition, Brazilian aspects are mentioned more than necessary, which is also true for the references. In general, the article does not add too much to the data in existing reviews of the topic, thus, it is less suitable for providing new information on the subject to the international professional community. Moreover, the text is not very well-written, there is a need for a thorough English grammar revision. Based on all these, the manuscript cannot be recommended for publication in Veterinary Sciences in its present form. It would be more appropriate for publication in a local professional journal.
Reviewer 4 Report
Please see the attachment

Round 2
Reviewer 3 Report
Although the authors have made some changes that improved the manuscript a bit, however, it is still far from the standard expected from an international literature review. Consequently, the manuscript is still not recommended for publication in Veterinary Sciences in its current form.
Comments:
Style and language: the standard of English still needs definite improvement. In many cases, the wording is confusing and complicated. The misspellings and linguistic errors of the previous version are still present in this version.
The title is not concise enough, it needs reformulation
Table 1. ORT is catalase negative (as stated in the cited literature as well).
Table 2 is redundant, it is sufficient to mention commercial identification systems in the text.
Figure 2 is also redundant, this information can be described in one sentence in the text.
Table 4 contains only partial data, the essence could be summarized in a few sentences in the text in brackets with the appropriate references.
Line 350: "Until now, the profile of susceptibility to antimicrobials has not been determined" This is not true, the fact is that there is no CLSI standard.
The title of citation 72 is incorrect: “Molecular characterization of a recently emerged poultry pathogen by multilocus sequence”. Correctly: "Molecular Characterization of Recently Emerged Poultry Pathogen Ornithobacterium Rhinotracheale by Multilocus Sequence Typing".
Author Response
Responses to Reviewer 3
We are very grateful by the important observations about our manuscript, which allow to improve their quality.
Point 1: Style and language: the standard of English still needs definite improvement. In many cases, the wording is confusing and complicated. The misspellings and linguistic errors of the previous version are still present in this version. Response 1: In addition to the corrections and suggestions made, we reevaluated the entire manuscript to make its text more homogeneous. The English writing was revised again.----------------------------------------------
Point 2: The title is not concise enough, it needs reformulation.
Response 2: We agree with the need for title reformulation and change it to a more concise form, as our manuscript explores various points of the ORT and not just its importance as an avian pathogen and disease strategy control.----------------------------------------------
Point 3: Table 1. ORT is catalase negative (as stated in the cited literature as well).
Response 3: Regarding Table 1, the reading correction of the catalase test result was made and we thank you for your attention in this review.----------------------------------------------
Point 4: Table 2 is redundant, it is sufficient to mention commercial identification systems in the text.
Response 4: As suggested, the Table 2 has been removed and mentioned in the form of textual sentences (Line 245 – 251).----------------------------------------------
Point 5: Figure 2 is also redundant, this information can be described in one sentence in the text.
Response 5: As suggested, the Figure 2 has been removed and the main idea was cited in the conclusion (Line: 431-415).
Point 6: Table 4 contains only partial data, the essence could be summarized in a few sentences in the text in brackets with the appropriate references. Response 6: As suggested, the Table 4 has been removed and its data inserted in a few sentences in the text with respective references (Line: 306-316).
Point 7: Line 350: "Until now, the profile of susceptibility to antimicrobials has not been determined" This is not true, the fact is that there is no CLSI standard.
Response 7: Regarding the determination of the antimicrobial susceptibility profile by CLSI, the sentence was clarified and the alterations in the writing were made (Line 299-301). -
Point 8: The title of citation 72 is incorrect: “Molecular characterization of a recently emerged poultry pathogen by multilocus sequence”. Correctly: "Molecular Characterization of Recently Emerged Poultry Pathogen Ornithobacterium rhinotracheale by Multilocus Sequence Typing".
Response 8: The title of citation 72 was corrected.
We hope to have complied the modifications that may allow this manuscript to be in agreement for publication in Veterinary Sciences.
Best regards,
Eunice Ventura
Round 3
Reviewer 3 Report
The authors responded satisfactorily to my concerns, so I now support publishing the manuscript.